# An open-source device for measuring food intake and operant behavior in rodent home-cages

**Bridget A Matikainen-Ankney[1†], Thomas Earnest[1†], Mohamed Ali[2,3†], Eric Casey[1], Justin G Wang[4], Amy K Sutton[2], Alex A Legaria[4], Kia M Barclay[4], Laura B Murdaugh[5], Makenzie R Norris[4,6], Yu-Hsuan Chang[4], Katrina P Nguyen[2], Eric Lin[1], Alex Reichenbach[7], Rachel E Clarke[7], Romana Stark[7], Sineadh M Conway[6,8], Filipe Carvalho[9], Ream Al-Hasani[6,8], Jordan G McCall[6,8], Meaghan C Creed[1,4,8], Victor Cazares[10], Matthew W Buczynski[5], Michael J Krashes[2], Zane B Andrews[7], Alexxai V Kravitz[1,4,8]***

[1]Department of Psychiatry, Washington University in St. Louis, St. Louis, United States; [2]National Institute of Diabetes and Digestive and Kidney Diseases, Bethesda, United States; [3]Department of Bioengineering, University of Maryland, College Park, United States; [4]Department of Neuroscience, Washington University in St. Louis, St. Louis, United States; [5]Department of Neuroscience, Virginia Polytechnic and State University, Blacksburg, United States; [6]Center for Clinical Pharmacology, University of Health Sciences and Pharmacy, St. Louis, United States; [7]Department of Physiology, Monash University, Clayton, Australia; [8]Department of Anesthesiology, Washington University in St. Louis, St. Louis, United States; [9]Open Ephys Production Site, Lisbon, Portugal; [10]Department of Psychology, Williams College, Williamstown, United States

***For correspondence:**
alexxai@wustl.edu

[†]These authors contributed equally to this work

**Abstract** Feeding is critical for survival, and disruption in the mechanisms that govern food intake underlies disorders such as obesity and anorexia nervosa. It is important to understand both food intake and food motivation to reveal mechanisms underlying feeding disorders. Operant behavioral testing can be used to measure the motivational component to feeding, but most food intake monitoring systems do not measure operant behavior. Here, we present a new solution for monitoring both food intake and motivation in rodent home-cages: the Feeding Experimentation Device version 3 (FED3). FED3 measures food intake and operant behavior in rodent home-cages, enabling longitudinal studies of feeding behavior with minimal experimenter intervention. It has a programmable output for synchronizing behavior with optogenetic stimulation or neural recordings. Finally, FED3 design files are open-source and freely available, allowing researchers to modify FED3 to suit their needs.

## Introduction

Feeding is critical for survival, and dysregulation of food intake underlies medical conditions such as obesity and anorexia nervosa. Quantifying food intake is necessary for understanding these disorders in animal models. However, it is challenging to accurately measure food intake in rodents due to the small volume of food that they eat. Researchers have devised multiple methods for quantifying food intake in rodents, each with advantages and drawbacks (*Ali and Kravitz, 2018*). Manual weighing of a food container is a simple and widely used method for quantifying food intake in a rodent home-cage. Yet this is time consuming to complete, is subject to error and variability, and

**eLife digest** Obesity and anorexia nervosa are two health conditions related to food intake. Researchers studying these disorders in animal models need to both measure food intake and assess behavioural factors: that is, why animals seek and consume food.

Measuring an animal's food intake is usually done by weighing food containers. However, this can be inaccurate due to the small amount of food that rodents eat. As for studying feeding motivation, this can involve calculating the number of times an animal presses a lever to receive a food pellet. These tests are typically conducted in hour-long sessions in temporary testing cages, called operant boxes. Yet, these tests only measure a brief period of a rodent's life. In addition, it takes rodents time to adjust to these foreign environments, which can introduce stress and may alter their feeding behaviour.

To address this, Matikainen-Ankney, Earnest, Ali et al. developed a device for monitoring food intake and feeding behaviours around the clock in rodent home cages with minimal experimenter intervention. This 'Feeding Experimentation Device' (FED3) features a pellet dispenser and two 'nose-poke' sensors to measure total food intake, as well as motivation for and learning about food rewards. The battery-powered, wire-free device fits in standard home cages, enabling long-term studies of feeding behaviour with minimal intervention from investigators and less stress on the animals. This means researchers can relate data to circadian rhythms and meal patterns, as Matikainen-Ankney did here.

Moreover, the device software is open-source so researchers can customise it to suit their experimental needs. It can also be programmed to synchronise with other instruments used in animal experiments, or across labs running the same behavioural tasks for multi-site studies. Used in this way, it could help improve reproducibility and reliability of results from such studies.

In summary, Matikainen-Ankney et al. have presented a new practical solution for studying food-related behaviours in mice and rats. Not only could the device be useful to researchers, it may also be suitable to use in educational settings such as teaching labs and classrooms.

does not allow for fine temporal analysis of consumption patterns (*Acosta-Rodríguez et al., 2017*; *Reinert et al., 2019*). Automated tools have been developed for measuring food intake in home-cages with high temporal resolution, although most require modified caging, powdered foods, or connected computers which limit throughput (*Ahloy-Dallaire et al., 2019*; *Farley et al., 2003*; *Moran, 2003*; *Yan et al., 2011*). These include automated weighing (*Hulsey and Martin, 1991*; *Meguid et al., 1990*; *Minematsu et al., 1991*), pellet dispensers (*Aponte et al., 2011*; *Gill et al., 1989*; *Oh et al., 2017*), or video detection-based systems (*Burnett et al., 2016*; *Jhuang et al., 2010*; *Salem et al., 2015*).

In addition to measuring total food intake, understanding neural circuits involved in feeding requires exploring *why* animals seek and consume food. Has their motivation for a specific nutrient changed? Has their feeding gained a compulsive nature that is insensitive to satiety signals? These questions can be answered with operant tasks, where rodents receive food contingent on their actions (*Curtis et al., 2019*; *Mourra et al., 2020*; *O'Connor et al., 2015*; *Skinner, 1938*; *Thorndike, 1898*; *Wald et al., 2020*). Typically, operant behavior is tested in dedicated chambers for a few hours each day. Commercial systems for testing operant behaviors typically comprise a dedicated arena equipped with levers, food or liquid dispensers, tones or spotlights, and video cameras. Programmable software interfaces allow experimenters to control this equipment to train animals on behavioral tasks such as fixed-ratio or progressive ratio responding. Some chambers are also equipped with video cameras and tracking software, so the location of the animal can be used as a trigger to control task events (*London et al., 2018*). Training in dedicated operant chambers has limitations: tasks can take weeks for animals to learn, animals may be tested at different phases of their circadian cycle due to equipment availability, and food restriction can be necessary to get animals to seek food outside of their home-cage, which can confound feeding studies. To mitigate these issues, several researchers have begun to test operant behavior in rodent home-cages, resulting in both fewer interventions from the researcher and faster rates of learning (*Balzani et al., 2018*; *Francis and Kanold, 2017*; *Lee et al., 2020*).

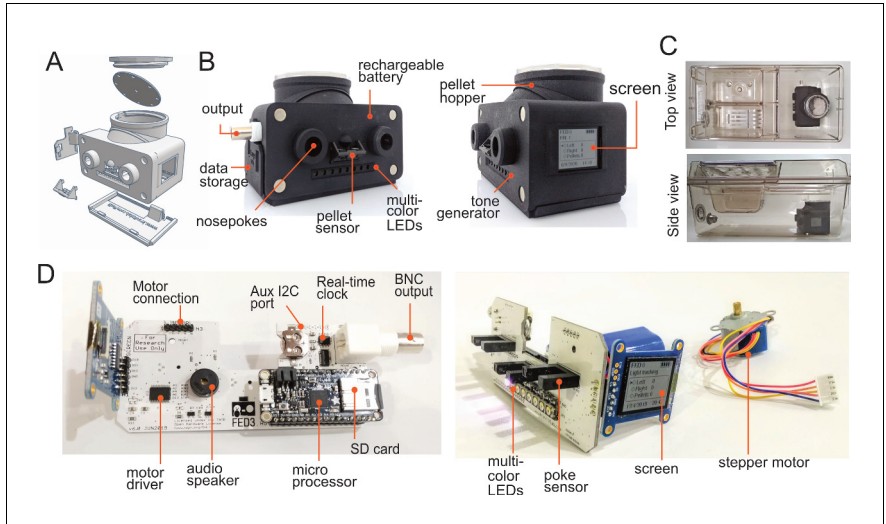

**Figure 1.** Assembly of FED3. (**A**) Exploded view schematic and (**B**) photos of FED3 with main components highlighted. (**C**) Assembled FED in an Allentown NextGen home-cage, top view, and side view. (**D**) Back view (left) of assembled and populated FED3 PCB, and side view (right) of assembled FED3 electronics.

The online version of this article includes the following figure supplement(s) for figure 1:

**Figure supplement 1.** FED3 circuit.

Here, we present a new solution for monitoring food intake and testing operant behavior in rodent home-cages: the Feeding Experimentation Device version 3 (FED3). Our goal was to develop a device for measuring food intake in rodent home-cages with high temporal resolution, while also measuring food motivation via operant behavior. FED3 is a stand-alone device that contains a pellet dispenser, two 'nose-poke' sensors for operant behavior, visual and auditory stimuli, and a screen for experimenter feedback. FED3 is compact and battery powered, fitting in most commercial vivarium home-cages without any connected computers or external wiring. FED3 also has a programmable output that can control other equipment, for example to trigger optogenetic stimulation after a nose-poke or pellet removal, or to synchronize feeding behavior with electrophysiological or fiber-photometry recordings, or with in vivo calcium imaging. Finally, FED3 is open source and was designed to be customized and re-programmed to perform novel tasks to help researchers understand food intake and food motivation. To this end, we have written a user-friendly Arduino library to facilitate custom behavioral programming of FED3. Here, we describe the design and construction of FED3 and present several experiments that demonstrate its functionality. These include measuring circadian patterns of food intake over multiple days, performing meal pattern analysis, automated operant training, closed-economy motivational testing, and optogenetic self-stimulation. FED3 extends existing methods for quantifying food intake and operant behavior in rodents to help researchers achieve a deeper understanding of feeding and feeding disorders.

## Results

### Hardware and design of the FED3

FED3 is controlled by a 48 mHz ATSAMD21G18 ARM Cortex M0 microprocessor (Adafruit Feather M0 Adalogger) and contains two nose-pokes, a pellet dispenser, a speaker for auditory stimuli, eight multi-color LEDs for visual stimuli, a screen for experimenter feedback, and a programmable analog output (*Figure 1A,B*). When mice interact with FED3, the timing of each poke and pellet removal are logged to an internal microSD card for later analysis and summary data is displayed on the screen for immediate feedback to the researcher (*Figure 1B*). FED3 is small (~10 cm × 12 cm × 9 cm), battery powered, and completely self-contained, so it fits in most vivarium home-cages without modification or introducing wiring to the cage (*Figure 1C*). It is powered by a rechargeable battery that lasts ~1 week between charges (exact battery life depends on the behavioral program being

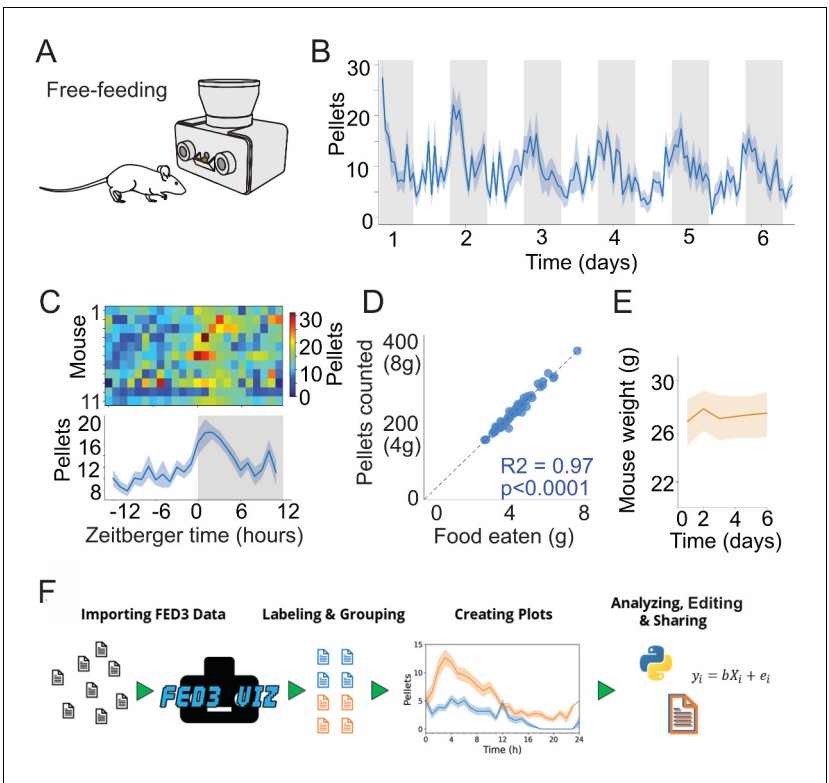

**Figure 2.** FED3 tracks food intake. (**A**) Schematic of FED3 in free-feeding mode. (**B**) Pellets per hour across six consecutive days. Shaded areas indicate dark cycle. (**C**) Chronograms of pellets eaten in heatmap (top) and line plot. (**D**) Regression of the calculated weight of the pellets recorded by FED vs. the measured difference in weight of the FED between days. (**E**) Mice weight across time during free feeding. (**F**) Schematic of FED3VIZ workflow. Data is shown as means ± SEM in (**B, C, E**).

The online version of this article includes the following figure supplement(s) for figure 2:

**Figure supplement 1.** FED3 pellet delivery cute.

run and number of pellets dispensed). FED3 also has magnetic mounts to facilitate wall-mounting on any plastic box to mimic a traditional operant setup (see *Video 1*). We provide example programs for running rodents on multiple common behavioral paradigms, including free feeding, time-restricted free-feeding, fixed-ratio 1 (FR1), and progressive ratio (PR) operant tasks, and have written a FED3 Arduino library to ease development of custom programs. FED3 also has an auxiliary I2C port for hardware customization and a programmable BNC output connection that allows synchronization with external equipment for aligning behavioral events with fiber-photometry recordings (*London et al., 2018*; *Mazzone et al., 2020*), electrophysiological recordings (*London et al., 2018*), or other equipment such as video tracking systems (*Krynitsky et al., 2020*; *Li et al., 2019*). This BNC output can report task events or control external hardware with delays of less than 200 µs, which compares favorably with computers that process information through Universal Serial Bus (USB), which can introduce delays of several milliseconds. Finally, we provide a graphical analysis package (FED3VIZ) written in Python that enables users to generate detailed plots from FED3 data (*Figure 2F*). FED3 is open-source and freely available online, including 3D design files, printed circuit board (PCB) files (*Figure 1—figure supplement 1A,B*), build instructions (*Figure 1D*), and code (https://github.com/KravitzLabDevices/FED3 copy archived at swh:1:rev:ff8fc79d288a440d91566f4c6cd80011956f4be0).

## Quantifying total food intake

Ten FED3 devices were set to the 'free-feeding' program, which dispenses a pellet and monitors its presence in the pellet well (*Figure 2A*). In this paradigm, each time the pellet is removed, FED3 logs

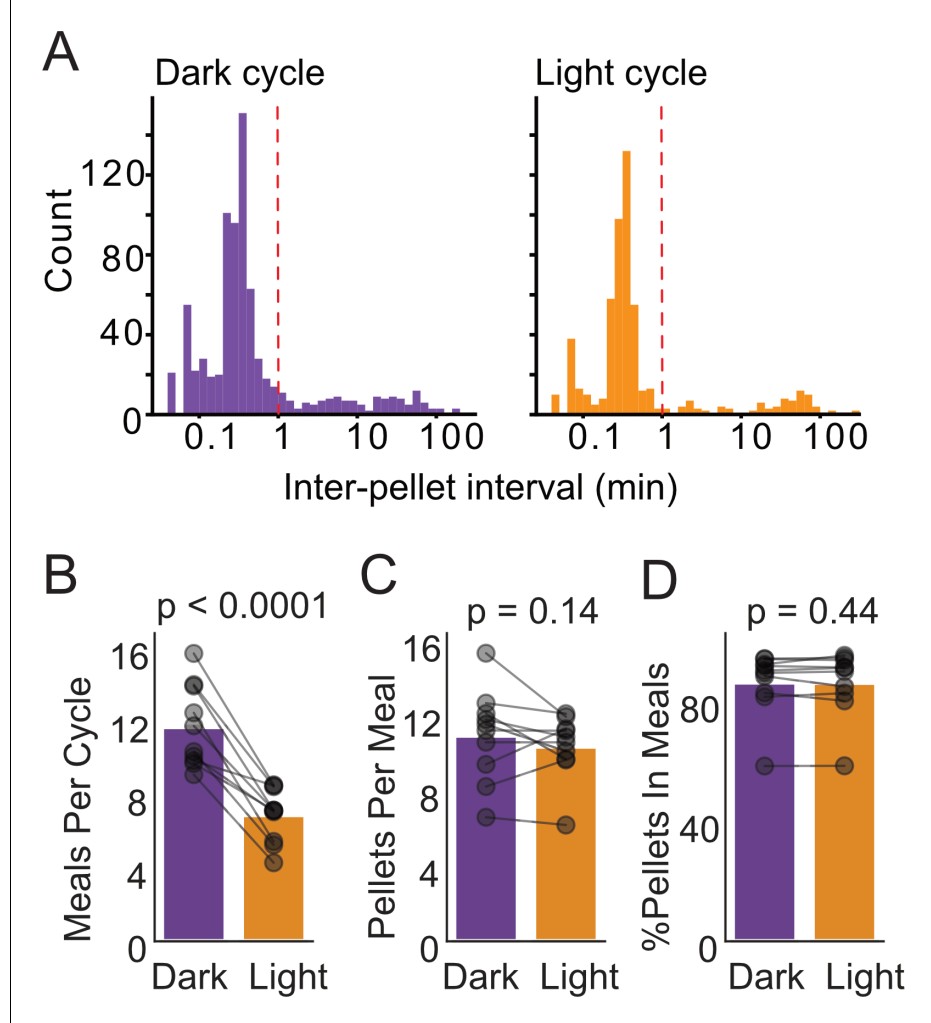

**Figure 3.** Meal analysis with FED3. (**A**) Inter-pellet interval histograms for dark and light cycle feeding. (**B**) Meals per day, (**C**) pellets per meal, and (**D**) % of pellets within meals for dark and light cycle feeding. N = 10 mice, paired t-tests.

the date and time to the storage card and dispenses another pellet. This allows for the reconstruction of detailed feeding records over multiple days. Assuming calorie content of the pellets is known, total caloric intake can also be tracked over time. Devices were placed with ten singly housed mice for six consecutive days with no additional food source. We observed the expected circadian rhythm in food intake (*Figure 2B,C*). To test accuracy, we confirmed that the daily change in weight of the FED3 device correlated with the number of pellets removed multiplied by the weight of each pellet (20 mg pellets, *Figure 2D*). The coefficient of determination ($R^2$) for the regression was 0.97, indicating an error rate of ~3% between manual weighing and counting of pellets by FED. Finally, we confirmed that the mice obtained all of their necessary daily calories from FED3, evidenced by a stable body weight across these 6 days (*Figure 2E*). In other experiments, we have run FED3 for >1 month without

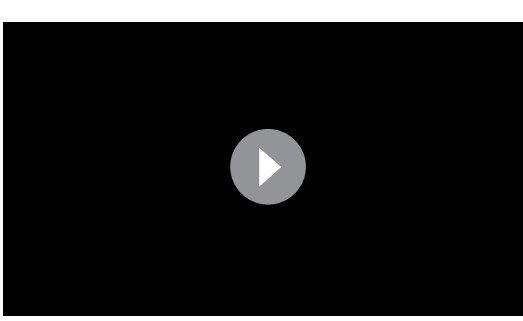

**Video 1.** Example movie of mouse using FED3 in a home-cage, and with FED3 mounted on a plastic box.
https://elifesciences.org/articles/66173#video1

observing weight loss. FED3 can be used for multi-week experiments like this as it has a low rate of jamming. This is facilitated by a unique angled pellet hopper and dispensing path that has fewer places where pellets can jam than traditional horizontal pellet dispensers (*Figure 2—figure supplement 1*), as well as 'jam clearing' movements in the code that vibrate and rotate the dispensing disk to dislodge stuck pellets when they are detected. These are improvements over the first generation of this device (*Nguyen et al., 2016*). As a point on rodent safety, although FED3 is resistant to jamming it still must be checked for function at least once per day when being used as the only food source.

FED3 creates a large amount of data, particularly when run over multiple days. To facilitate analysis of FED3 data, we created FED3VIZ, an open-source Python-based analysis program (https://github.com/earnestt1234/FED3_Viz). FED3VIZ offers plots for visualizing different aspects of FED3 data, including pellet retrieval, poke accuracy, pellet retrieval time, delay between consecutive pellet earnings (inter pellet intervals), meal size, and progressive ratio breakpoint. These metrics can be plotted for single files or group averages. FED3VIZ's averaging methods provide options for aggregating data recorded on different days, while preserving time-of day or phase of the light–dark cycle. Furthermore, circadian activity patterns can also be visualized with the FED3VIZ 'Chronogram' and 'Day/Night Plot' functions (see Figure 7), which segment data based on a user-defined light cycle. The processed data going into each plot can also be saved and used to create new visualizations or compute statistics in other programs, promoting reproducible, sharable analysis and visualization of FED3 data.

## Meal analysis

FED3 records the date and time that each pellet is removed, which can be used to quantify feeding patterns including the size and quantity of meals, eating rate within meals, and timing between meals. Different parameters have been used by different groups to define meals, often based on numerical cut-offs (*Farley et al., 2003*; *Kanoski et al., 2013*; *Melhorn et al., 2010*). To complement

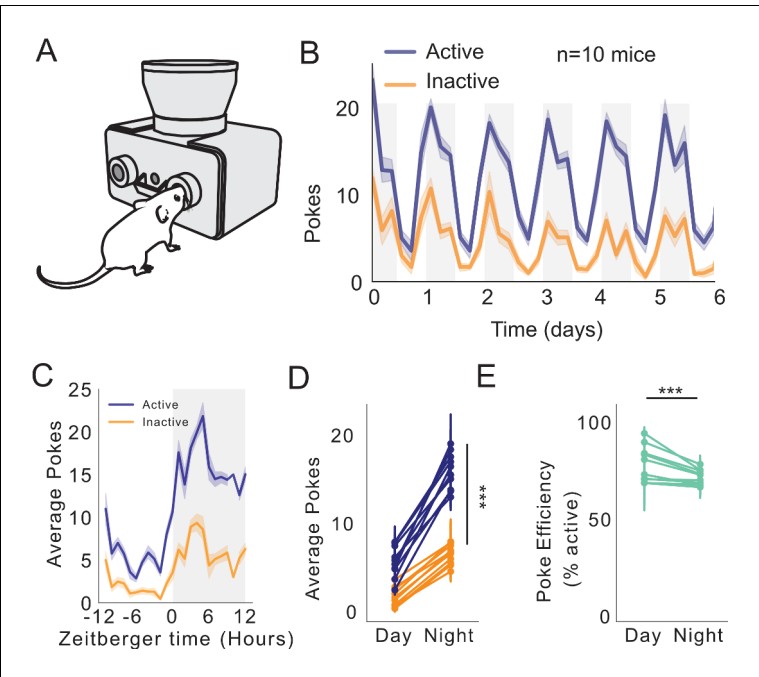

**Figure 4.** FED3 reveals circadian feeding patterns. (A) Schematic of FED3 in FR1 mode. (B) Active (blue) and inactive (orange) pokes over 6 days. n = 10 mice. (C) Average pokes (active, blue; inactive, orange) over 24 hr cycle. (D) Average pokes during light and dark cycles. Significant interaction between day/night and average pokes ($F_{(1,792)}=85.225$, $p<0.0001$); significant effect of day/night ($F_{(1,792)}=668.700$, $p<0.0001$); significant effect of active or inactive pokes ($F_{(1,792)}=610.824$, $p<0.0001$). Fitted linear model for two-way ANOVA, ($F_{(3,792)}=454.917$, $p<0.0001$.) (E) Average poke efficiency ($p=0.0001$, student's t-test).

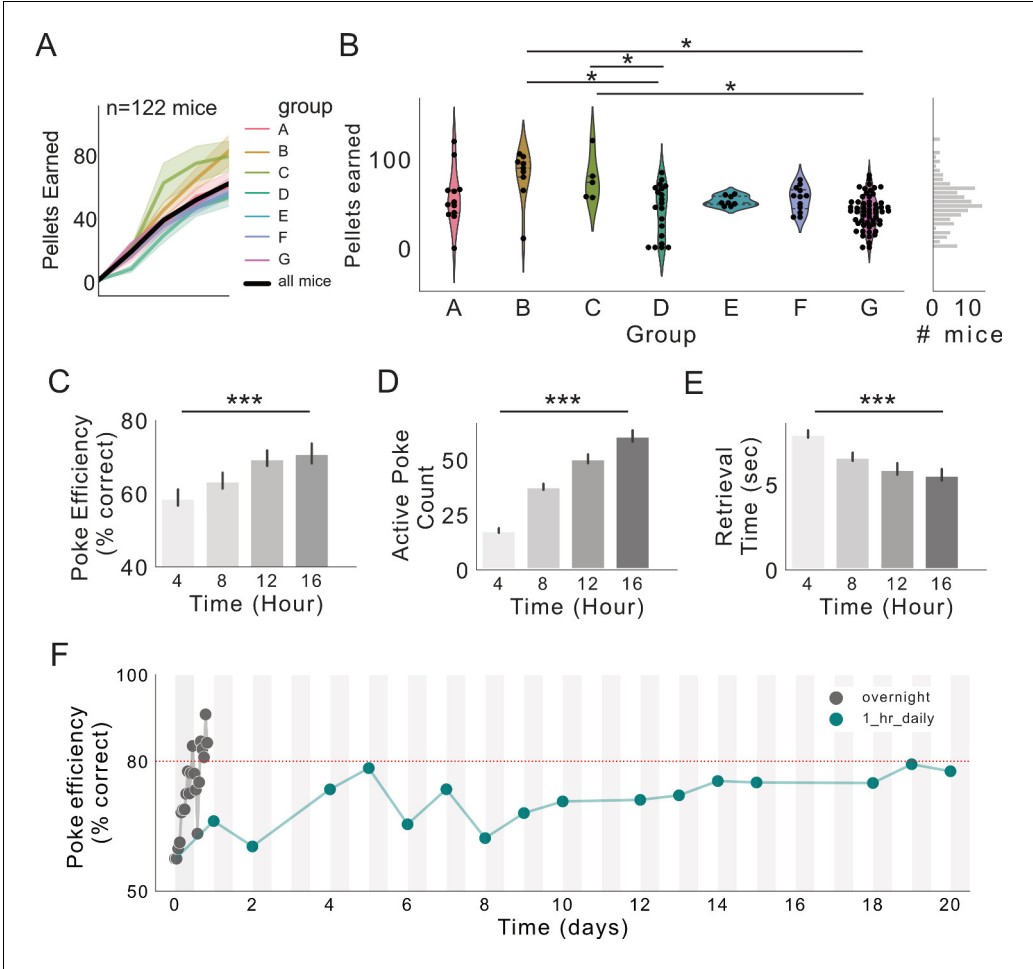

**Figure 5.** FR1 acquisition across seven research sites. (**A**) Pellets earned over first 16 hr of exposure to FED3 at seven research sites. (**B**) Scatterplots and kernel density estimation plots showing pellets earned after 16 hr with FED3, effect of group ($F_{(6,115)}$=6.4223, p<0.0001), significant post hoc differences between groups B and D (p=0.001), B and G (p=0.001), C and D (p=0.030), and C and G (p=0.010). (**C**) Poke efficiency across the session. Significant effect of time, p=0.0007, F(3,386)=5.79. (**D**) Active pokes across the session. Significant effect of time, p=0.0001, F(3, 393)=115.2. (**E**) Retrieval time across the session. Significant effect of time, p=0.0001, F(3,351) =14.02. (**E**) Linear models with Tukey post-tests were conducted in (**C-E**). (**F**) Poke efficiency across continuous 16 hr sessions (gray line, n = 122 mice) and across multiple days with 1 hr sessions each day (teal line, n = 11 mice). Two-way ANOVA revealed significant effect of time (p<0.0001), no significant effect of group or interaction. F (7,500)=3.974.

these approaches, we aimed to develop an unbiased approach for understanding meal patterns, based on the distribution of time intervals between each pellet consumed. For the 10 mice in the above experiment, we plotted the inter-pellet time interval histogram (*Figure 3A*) for both the dark and light cycles, based on an approach that was previously described in rats (*Cottone et al., 2007*). We observed a large peak at <1 min between pellets, showing that the majority of pellets were consumed within 1 min of another pellet. We used this distribution to classify pellets eaten within the same minute as belonging to the same meal, and pellets with larger intervals between them belonging to different meals. This approach revealed that animals eat fewer meals during the light cycle (*Figure 3B*) but eat similar number of pellets within each meal and obtain a similar fraction of their pellets within meals in each cycle. Differences in meal patterning have been linked to obesity (*Farley et al., 2003*; *Wald and Grill, 2019*), so this analytical approach may assist in understanding obesity and other disorders of feeding.

## Circadian patterns of operant behavior

A unique feature of FED3 is that it is small and wire-free, allowing it to fit inside of traditional vivarium home-cages. This facilitates quantification of circadian rhythms in both operant and feeding behavior (*Figure 4A*). To demonstrate operant capability, we quantified how nose-poking for pellets varied over the circadian cycle, by running mice on a FR1 task for six consecutive days. Mice learned to nose-poke on the active port for a single pellet. We tracked active (rewarded) and inactive (unrewarded) pokes. As expected, both active and inactive pokes were higher during the dark cycle (*Figure 4B–D*). Surprisingly, however, poke *efficiency* (% of total pokes on the active port) was slightly higher during the light cycle (*Figure 4E*), suggesting that operant responding is more efficient during the light cycle.

## A multi-site study of learning rates with FED3

Multiple research groups currently use FED3. We therefore asked how mouse operant learning varies across different laboratories. To do this, we obtained operant data from the first overnight FR1 session from seven laboratories, resulting in data from 122 mice (mix of males and females, *Figure 5A*). When comparing pellets earned in an overnight session across laboratories, we observed a significant effect of group, linked to significant differences in 4 of 21 post hoc comparisons (*Figure 5B*). When viewed as a single distribution (*Figure 5B* inset), the distribution of pellets earned from all groups was consistent with a single Gaussian distribution (p>0.05, Shapiro-Wilk test for normality). This highlights how FED3 enables high-throughput studies of operant behavior, and the power of large sample sizes to achieve adequate sampling of a distribution of behavior (*Figure 5B*).

A unique capability of FED3 is that it has a sensor for detecting the presence of the pellet in the feeding well. This enables time-stamping of when each pellet was removed for constructing feeding records, as well as measuring the time between the pellet dispense and removal, which we termed 'retrieval time'. Changes in retrieval time may reflect learning, as this measure progressively decreased as animals gained experience with FED3 (*Figure 5E*). Accordingly, active active count and poke efficiency (% of pokes on the active port) increased over time, demonstrating that retrieval time decreased on the same time scale that mice learned which poke resulted in a pellet (*Figure 5C*). To compare learning rates in overnight training to traditional daily operant sessions, we ran a new group of eleven mice for sixteen daily 1 hr FR1 sessions with FED3, returning them to the colony for the remainder of each day. Learning rates did not differ between mice exposed to 16 daily 1 hr sessions or one 16 hr overnight session (*Figure 5F*). This suggests that the acquisition of this FR1 task may depend most strongly on the cumulative time mice interact with FED3 and that overnight training can greatly speed up task acquisition by giving them many hours of experience in one night.

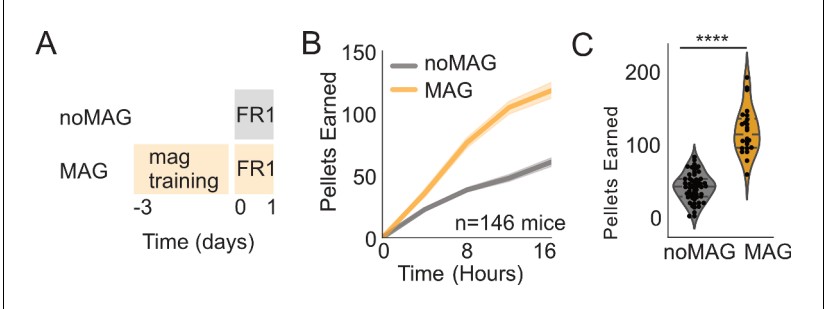

**Figure 6.** Effect of magazine training. (**A**) Schematic showing paradigm for magazine trained group (MAG) vs. no magazine training (noMAG). (**B**) Active pokes over time during first exposure to FED3 FR1 between noMAG and MAG groups. (**C**) Scatterplots and kernel density estimation plots showing distribution of active poke counts at 16 hr (p=0.0001). Student's t-test. N = 146 mice.

## The effect of magazine training on acquisition of operant behavior

Based on prior literature (*Steinhauer et al., 1976*), we predicted that the speed of acquisition of an FR1 task would be further increased by prior magazine training. To test this, we exposed 24 mice to magazine training using the free feeding FED3 paradigm for 1–3 days prior to FR1 training (*Figure 6A*). This paradigm dispenses a pellet into the feeding well of FED3 and replaces it whenever it is removed, providing an automatic method for magazine training animals. In this way, magazine training allows mice to learn the location of the food source and associate food with the sound of the pellet dispenser operating and the pellet being dispensed, before commencing with FR1 training. We compared the performance of the 24 magazine trained mice to 122 mice from *Figure 5* (which had not been magazine trained) and found that magazine training resulted in significantly higher (~2) levels of pellet acquisition in the first night with the FR1 task (*Figure 6B,C*). Therefore, magazine training is recommended to speed up acquisition of nose-poking tasks.

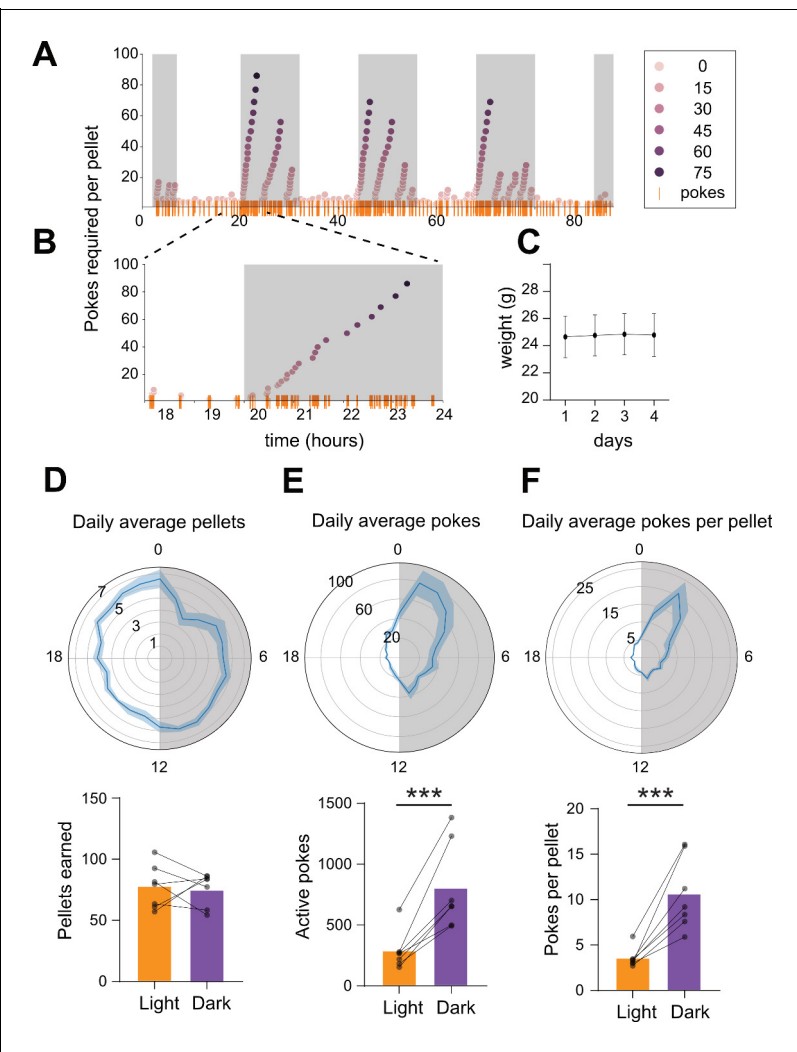

**Figure 7.** Longitudinal closed-economy feeding with FED3. (A) Active pokes required per pellet in a 4 day closed-economy progressive ratio task with 30 min resets. (B) Inset illustrating increasing number of pokes (y-axis, and orange rasters) per pellet earned during dark cycle. (C) Mouse weights over time (n = 7 mice). (D–F) Chronograms (top) and bar/scatter plots (bottom) showing daily average pellets (D), pokes (E), and pokes per pellet (F) over time, binned by 1 hr. Shaded region in chronograms indicates dark cycle. Significant increase in active pokes, p=0.0044 (E, bottom), and pokes per pellet, p=0.0007 (F, bottom) during dark cycle. No significant difference in pellets earned during dark cycle, p=0.8622 (D, bottom).

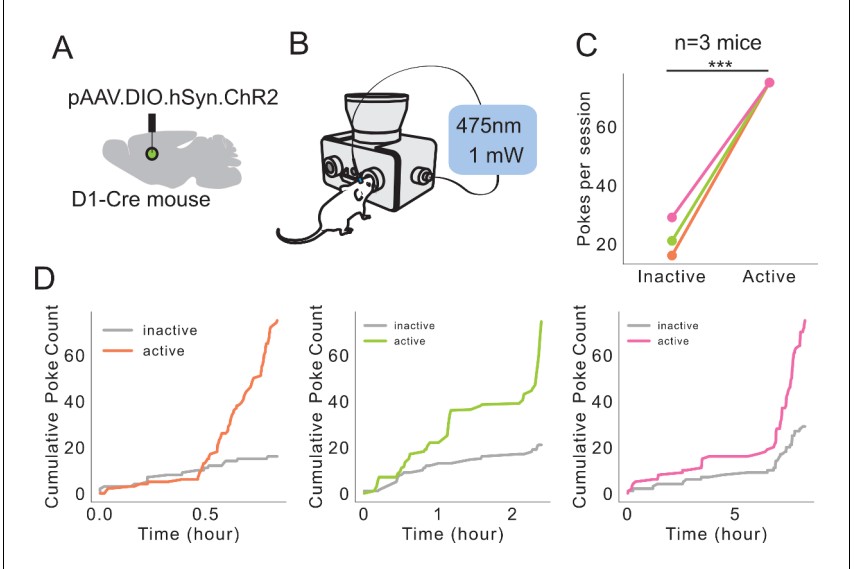

**Figure 8.** Self-stimulation of dMSNs using FED3. (**A**) Schematic showing Cre-dependent ChR2 injected into dorsal medial striatum with fiber optic implanted. (**B**) Schematic showing FED3 operant self-stim setup. (**C**) Mice poked significantly more on the active port (p=0.0002, Student's t-test, n = 3 mice). (**D**) Cumulative inactive and active pokes plotted over time for each mouse.

## Closed-economy food intake with FED3

FED3 can also be used for closed-economy feeding studies, in which all food is earned by operant nose-poking (*Figure 7*). Closed-economy operant tasks have been used to quantify the economic demand for food (*Bauman, 1991*; *Chaney and Rowland, 2008*), as well as changes in economic demand due to manipulation of the dopamine system (*Beeler et al., 2010*; *Mourra et al., 2020*). We tested seven mice on a repeating progressive ratio task (adapted from *Mourra et al., 2020*), in which the nose-poking requirement began on FR1 and increased each time a pellet was earned. When mice refrained from poking on either port for 30 min the ratio reset to FR1 (*Figure 7A,B*). Although pellets earned from this task were their only food source, mouse weights remained stable across days, demonstrating that mice earned their entire caloric need with the task (*Figure 7C*). Total pellets earned did not differ between light and dark cycles (*Figure 7D*). However, nose-poking rates were higher during the dark phase of their daily cycle (*Figure 7E*), meaning mice performed more pokes to earn each pellet in this phase (*Figure 7F*). This suggests that motivation for food pellets varies over the circadian cycle such that mice are willing to work harder during the dark cycle vs. the light cycle. When the data was visualized hourly across the circadian cycle, we further noted a ~3 hr window of enhanced nose-poking rates starting shortly after the onset of the dark cycle (*Figure 7F*). This time coincides with a circadian peak in extracellular dopamine levels in the striatum (*Ferris et al., 2014*), and a peak in activity levels in mice (*Matikainen-Ankney et al., 2020*). Therefore, this task may be used to assay changes in accumbal dopamine function across the circadian cycle.

## Optogenetic self-stimulation with FED3

FED3 was also designed to synchronize with other experimental equipment through its programmable output port. To demonstrate this capability, we programmed FED3 to act as a function generator and control an LED for an intra-cranial self-stimulation task. The dorsal striatum contains two main classes of output neurons, differentiated by their expression of the dopamine D1 or D2 receptors (*Gerfen et al., 1990*). Dorsal striatal neurons that express the D1 receptor are highly reinforcing when optogenetically stimulated (*Kravitz et al., 2012*). To demonstrate how FED3 can be used to perform an optogenetic self-stimulation experiment, we used Cre-dependent viral expression to target channelrhodopsin-2 (ChR2) to D1R-expressing neurons in the dorsal striatum (*Figure 8A*). FED3 was programmed to generate and trigger a 1 s train of 20 Hz pulses of 475 nm light upon each

active nose-poke, and the session continued until they received 75 trains of stimulation (*Figure 8B*). Three mice were run on this task, resulting in significantly greater active vs inactive pokes (*Figure 8C,D*), and demonstrating that FED3 can be used for optogenetic self-stimulation studies. The programmable output is attached to an on-board digital-to-analog converter chip, meaning that it can be programmed to output pulses of any voltage from 0 to 3 V, allowing FED3 to represent multiple behavioral events with one output line. The timing of the BNC output port is regulated by the SAMD21 microprocessor that has a documented pulsing precision of <1 μs.

## Discussion

Quantification of food intake is necessary to understand animal models of feeding disorders, as well as other medical conditions. To quantify food intake in rodent home-cages, we previously published the FED version 1, an open-source, stand-alone feeding device that can fit in a rack-mounted home-cage (*Nguyen et al., 2016*). FED records a time series of pellets removed, which can be used to reconstruct feeding records. FED has been used by multiple research groups to understand feeding (*Brierley et al., 2020*; *Burnett et al., 2019*; *Chen et al., 2020*; *Li et al., 2019*; *London et al., 2018*). Since its original publication, we have redesigned the FED twice and here present the third version of FED, which revamps our original design and adds the ability to measure operant behavior, as well as a unique 'angled' pellet dispenser mechanism that is resistant to jamming, which enables long term studies. We distributed the design for FED3 online, and it has been used to study how specific neural circuit manipulations alter food motivation (*Mazzone et al., 2020*; *Sciolino et al., 2019*; *Vachez et al., 2021*), how weight-loss alters food motivation (*Matikainen-Ankney et al., 2020*), and how food motivation is altered in a stress-susceptible mouse population (*Rodriguez et al., 2020*).

FED3 is a stand-alone solution for home-cage operant training, enabling researchers to understand not just total food intake, but also motivation for and learning about food rewards. FED3 has several unique benefits over comparable systems including: (1) FED3 is low cost. The FED3 electronics cost ~$200 and the housing is 3D printed. Even if 3D printing is done professionally this is >10× cheaper than most commercial solutions for measuring food intake or testing operant behavior; (2) FED3 is self-contained and fits within traditional vivarium caging, allowing for measures of true 'home-cage' feeding without modifying the cage. Due to its small size, it can also be placed inside of other equipment where wires might be impractical, such as within an indirect calorimetry system; (3) FED3 has a programmable output that allows it to easily synchronize with other equipment. Due to wiring, connecting FED3 to external hardware likely requires FED3 to be used outside of the home-cage, or to modify the home-cage system. This has been done by multiple labs to synchronize the output of FED3 with fiber-photometry recordings (*London et al., 2018*; *Mazzone et al., 2020*), electrophysiological recordings (*London et al., 2018*), optogenetic stimulation (*Vachez et al., 2021*), and video tracking (*Krynitsky et al., 2020*; *Li et al., 2019*); and (4) FED3 is open source and all design files and codes are freely available online. This enables users to modify the code and hardware to achieve new functionality.

The purpose of this manuscript is to demonstrate the utility of FED3 for feeding research. To this end, we demonstrate experiments that measured total food intake, operant responding, and optogenetic stimulation. We further highlight how the high temporal resolution enables meal pattern analysis across multiple days. Finally, we coordinated with six other research groups to compile a dataset of 122 mice across seven research sites, all running the same experimental FR1 program. We observed similar patterns of acquisition across all sites, demonstrating that FED3 can be used for multi-site research studies on feeding. Due to its low cost and open-source nature, we believe that multi-site studies with FED3 are more feasible than with other systems.

While FED3 has many strengths, it also has limitations. One limitation is that FED3 uses internal microSD cards to store data. While microSD cards are convenient, they are not ideal for large numbers of devices, where removing multiple cards can be cumbersome. Wireless data logging is a potential solution to this problem, although there are challenges to implementing this in rodent cages. A second limitation is that animals can 'hoard' pellets from FED3, as animals can remove food without consuming it. In our experience, this seemed to be a rare trait that specific mice engaged in (<10% of mice hoard pellets). Unfortunately, FED3 has no way to determine whether a mouse actually consumes each pellet after removal so we recommend checking for pellet hoarding and accounting for this in experimental conclusions. A third limitation is that granular bedding can

be kicked into the pellet well and interfere with pellet detection. To avoid this, we recommend using 'iso-pad' bedding, or bedding pellets that are large enough to avoid this possibility. A fourth limitation is that FED3 is not waterproof and would not be resistant to flooding, though the 3D printed housing is sufficient to protect electronics from typical wear associated with placement in a rodent cage. A final limitation of FED3 is that it has no way of identifying individual mice in group housed environments, so feeding records must be collected in singly housed mice. However, FED3 is open source, so it can be modified and improved with innovations from our group and the feeding community, and future iterations of the device may include methods for identifying multiple mice using radio-frequency identification (RFID) tags. Being able to run studies on group housed mice would greatly increase throughput and allow for the study of interactions between social behavior and feeding. We published the FED3 design as open-source and look forward to community contributions and modifications to enable new functionality and overcome these limitations.

# Materials and methods

## Key resources table

| Reagent type (species) or resource | Designation | Source or reference | Identifiers | Additional information |
|---|---|---|---|---|
| Strain, strain background (*Mus musculus*) | *Mus musculus* with name C57BL/6J from IMSR. | https://www.jax.org/strain/000664 | RRID:IMSR_JAX:000664 | |
| Software, algorithm | Python 3.7 (Anaconda Distribution) | https://www.anaconda.com/ | | |
| Software, algorithm | Spyder 4.1 | https://github.com/spyder-ide/spyder/releases | RRID:SCR_017585, version 4.1 | |
| Software, algorithm | Arduino IDE 1.8.1 | https://www.arduino.cc/ | | |
| Software, algorithm | FED3_VIZ | https://github.com/earnestt1234/FED3_Viz | | Analysis package for FED3 data files |
| Software, algorithm | TinkerCAD | https://www.tinkercad.com/ | | |
| Other | FED3 – commercially assembled | https://open-ephys.org/fed3/fed3 | | |
| Other | FED3 – open-source | https://github.com/KravitzLabDevices/FED3 | | |
| Other | 5TUM grain-based rodent enrichment pellets | https://www.testdiet.com/Diet-Enrichment-Products/Lab-Treat-Tablets-and-Pellets/index.html | | |

## Data and code availability

The FED3 device is open-source, and design files and code are freely available online at: https://github.com/KravitzLabDevices/FED3. In addition, we have made all data and analysis code for the figures in this paper available at https://osf.io/hwxgv/.

## Subjects

One hundred and fifty-nine C57Bl/6 mice were housed in a 12 hr light/dark cycle with ad libitum access to food and water except where described. Mice were provided laboratory chow diet (5001 Rodent Diet; Lab Supply, Fort Worth, TX). All procedures were approved by the Animal Care and Use Committee at Washington University in St Louis, the National Institutes of Health, Williams College, Virginia Tech, and Monash University.

## Design and construction of FED3

Tutorial videos and other information on assembling FED3 are available at: https://github.com/KravitzLabDevices/FED3/wiki.

### 3D design

The 3D parts for FED3 were designed with TinkerCAD (Autodesk). We have printed FED3 with an FDM printer in PLA (Sindoh 3DWox 1), as well as with a commercial SLS process in Nylon 12 (Shapeways). 3D files may need to be tweaked to obtain good results on different printers, so we provide editable design files here: https://www.tinkercad.com/things/0QaiVw7KR3Y.

### Electronics

Electronics design was completed using Autodesk Eagle version 9.3. FED3 is controlled by a commercial microcontroller (Adalogger Feather M0, Adafruit). This microcontroller contains an ATSAMD21G18 ARM M0 processor that runs at 48 MHz, 256 kB of FLASH memory, 32 kB of RAM memory, an on-board microSD card slot for writing data, and up to 20 digital inputs/output pins for controlling other hardware. The microcontroller also contains a battery charging circuit for charging the internal 4400mAhr LiPo battery in FED3, which provides ~1 week of run-time between charges. Exact battery life depends on the behavioral program and how often the mouse interacts with FED3. Additional hardware on FED3 includes a motor for controlling the pellet delivery hopper, eight multi-color LEDs for delivering visual stimuli, a small speaker for delivering audio stimuli, a screen for user feedback, and three infra-red beam-break sensors. Two of these sensors are used as 'nose-poke' sensors to determine when the mouse pokes, while the third is used to detect the presence and removal of each food pellet. Finally, FED3 contains a programmable output connected to a 10-bit digital-analog converter circuit, enabling full analog output of arbitrary voltages from 0 to 3.3 V. This can be used to synchronize FED3 with external equipment via digital pulses or analog signals. There are two variants of the electronics for FED3: An older design is referred to as the 'DIY FED3', which can be assembled by hand using commercially available components (https://hackaday.io/project/106885-feeding-experimentation-device-3-fed3), whereas the newer design contains more streamlined electronics but requires professional assembly. Both designs run the same code and have identical functionality.

### Firmware

The code for FED3 is written in the Arduino language and is fully open-source. The example code that we provide with FED3 contains several programs covering multiple common behavioral paradigms including free feeding, time-restricted free-feeding, FR1, and PR operant tasks, as well as an optogenetic self-stimulation mode that delivers trains of pulses for controlling a stimulation system. We also provide an Arduino library that simplifies control of FED3 hardware to simplify writing of custom programs.

## Behavioral testing with FED3

We demonstrate multiple operational modes for FED3 to highlight a range of functionality and applications. The following programs are included with the standard FED3 code:

### Free-feeding

In the free-feeding mode (*Figure 2*), a 20 mg pellet is dispensed into the feeding well and monitored with a beam-break. When the pellet is removed, the time-stamp of removal, and the latency to retrieve the pellet are logged to the internal microSD card and a new pellet is dispensed. For the free-feeding data in this paper, 10 mice were singly housed and the FED3 device was placed in their cage for 6 days on a 12/12 on/off light cycle. FED3 devices were checked for functionality each day, but mice were otherwise undisturbed.

### Fixed-ratio 1

In FR1 mode, FED3 logs the time-stamps of each 'nose-poke' event to the internal storage. When the mouse activates the left nose-poke, the FED3 delivers a combined auditory tone (4 kHz for 0.3 s) and visual (all eight LEDs light in blue) stimulus and dispenses a pellet. While the pellet remains in the well both pokes remain inactive to prohibit multiple pellets piling up in the well. When the pellet is removed, the time-stamp of removal, and the latency to retrieve the pellet are logged. For the data in this paper, the same 10 mice that completed the ree-feeding experiment were transitioned to an FR1 program, and FR1 data was recorded for an additional 6 days.

## Optogenetic stimulation

Viral infections of male and female Drd1-Cre mice were conducted under 0.5–2.5% anesthesia on a stereotaxic apparatus. Using a Nanoject injector, 500 nL of AAV2-DIO-hSyn-ChR2 was injected bilaterally (500 nL/hemisphere) in the dorsomedial striatum. Optical fibers were then implanted in the same region. After letting ChR2 express for 4 weeks, animals were pre-trained on an FR1 schedule for pellets overnight in their home-cage. Two days later, mice were placed in a box with a FED3 device connected to an LED driver. The active poke for this paradigm was on the opposite side to the active poke of the pre-training session. During the self-stimulation session, when the mouse poked on the active nose-poke, it received a 1 s train of 1 mW 475 nm light at 20 Hz. Sessions were run for 60 min in a 550 cm$^2$ arena.

## Data analysis

CSV files generated by FED3 were processed and plotted with custom python scripts (Python, version 3.6.7, Python Software Foundation, Wilmington, DE). All data and scripts are available on Open Science Framework (https://osf.io/hwxgv/). Visualization was also completed using FED3VIZ GUI to generate plots. FED3VIZ was written in Python's standard library for developing GUIs (tkinter). FED3-VIZ is a custom open-source graphical program for analyzing FED3 data. FED3VIZ code, version history, installation instructions, and user manual are available on GitHub (https://github.com/earnestt1234/FED3_Viz). FED3VIZ offers plotting and data output for visualizing different aspects of FED3 data, including pellet retrieval, poke accuracy, pellet retrieval time, delay between consecutive pellet earnings (inter-pellet intervals), meal size, and progressive ratio breakpoint. Based on inter-pellet interval histograms (*Figure 3*), we defined meals as pellets eaten within 1 min of each other. In addition, we defined a minimum size of 0.1 g (five pellets) to be counted as a meal.

## Statistics

Bartlett's test for equal variances was performed; one- or two-way ANOVAs with Tukey post-tests were used to compare groups with equal variance (p>0.05) where appropriate. Linear mixed effects models were used to analyze groups with significantly difference variances (p<0.05). p-value<0.05 was considered significant. Statistical tests to compare means were run using statsmodels module in python (*Seabold and Perktold, 2010*). Data sets are presented as mean ± SEM. Numbers of animals per experiment is listed as n=number of animals. Linear regression was used to determine correlative relationships. T-tests or Mann–Whitney U tests were used to compare the means of two groups for parametric or nonparametric data distributions, respectively.

## Acknowledgements

Funding was provided from the National Institutes of Health Intramural Research Program at NIDDK (AVK), the Washington University Diabetes Research Center Pilot and Feasibility Grant (AVK), the Washington University Nutrition Obesity Research Center (NORC) Pilot and Feasibility Program (AVK), the National Institute on Drug Abuse R00DA038725 (RA), the National Institute of Neurological Disorders and Stroke R01NS117899 (JGM), the McDonnell Center for Cellular and Molecular Neurobiology Postdoctoral Fellowship (BAMA), the National Health and Medical Research Council of Australia (ZBA), and the Imaging, Modeling and Engineering of Diabetic Tissues T32, NIDDK, DK108742 (BAMA), Brain and Behavior Research Foundation (NARSAD Young Investigator Grant 27416 to AVK and 27197 to MCC), National Institutes of Health National Institute on Drug Abuse (R21-DA047127, R01-DA049924 to MCC), Whitehall Foundation Grant (2017-12-54 to MCC) and Rita Allen Scholar Award in Pain (to MCC).

## Additional information

### Competing interests

Filipe Carvalho: Director of Open Ephys Production Site. The other authors declare that no competing interests exist.

## Funding

| Funder | Grant reference number | Author |
| --- | --- | --- |
| National Institutes of Health | ZIA DK075099 | Bridget A Matikainen-Ankney<br>Mohamed Ali<br>Amy K Sutton<br>Katrina P Nguyen<br>Michael J Krashes<br>Alexxai V Kravitz |
| Washington University Diabetes Research Center | | Bridget A Matikainen-Ankney<br>Thomas Earnest<br>Eric Casey<br>Alex A Legaria<br>Alexxai V Kravitz |
| Washington University Nutrition Obesity Research Center | | Kia M Barclay<br>Makenzie R Norris<br>Yu-Hsuan Chang<br>Meaghan C Creed |
| National Institutes of Health | R00DA038725 | Ream Al-Hasani |
| National Institutes of Health | R01NS117899 | Jordan G McCall |
| McDonnell Center for Systems Neuroscience | | Bridget A Matikainen-Ankney |
| National Health and Medical Research Council | | Zane B Andrews |
| National Institutes of Health | DK108742 | Bridget A Matikainen-Ankney |
| National Institutes of Health | DA047127 | Meaghan C Creed |
| National Institutes of Health | DA049924 | Meaghan C Creed |
| Whitehall Foundation | 2017-12-54 | Meaghan C Creed |
| Rita Allen Foundation | | Meaghan C Creed |
| Brain and Behavior Research Foundation | | Meaghan C Creed<br>Alexxai V Kravitz |
| National Institutes of Health | Intramural Research Program | Alexxai V Kravitz |
| Brain and Behavior Research Foundation | 27416 | Alexxai V Kravitz |
| Brain and Behavior Research Foundation | 27197 | Meaghan C Creed |

The funders had no role in study design, data collection and interpretation, or the decision to submit the work for publication.

## Author contributions

Bridget A Matikainen-Ankney, Data curation, Formal analysis, Visualization, Methodology, Writing - original draft, Writing - review and editing; Thomas Earnest, Mohamed Ali, Software, Methodology; Eric Casey, Validation, Investigation; Justin G Wang, Victor Cazares, Michael J Krashes, Investigation, Writing - review and editing; Amy K Sutton, Alex A Legaria, Kia M Barclay, Laura B Murdaugh, Makenzie R Norris, Yu-Hsuan Chang, Katrina P Nguyen, Alex Reichenbach, Rachel E Clarke, Romana Stark, Sineadh M Conway, Ream Al-Hasani, Jordan G McCall, Matthew W Buczynski, Investigation; Eric Lin, Filipe Carvalho, Methodology; Meaghan C Creed, Conceptualization, Investigation; Zane B Andrews, Supervision, Project administration; Alexxai V Kravitz, Conceptualization, Resources, Data curation, Software, Formal analysis, Supervision, Funding acquisition, Validation, Investigation, Visualization, Methodology, Writing - original draft, Project administration, Writing - review and editing

## Author ORCIDs

Eric Casey (ID) http://orcid.org/0000-0003-0041-6586
Ream Al-Hasani (ID) http://orcid.org/0000-0002-8781-6234

Jordan G McCall http://orcid.org/0000-0001-8295-0664
Michael J Krashes http://orcid.org/0000-0003-0966-3401
Zane B Andrews http://orcid.org/0000-0002-9097-7944
Alexxai V Kravitz https://orcid.org/0000-0001-5983-0218

## Ethics

Animal experimentation: 166 C57Bl/6 mice were housed in a 12-hour light/dark cycle with ad libitum access to food and water except where described. Mice were provided laboratory chow diet (5001 Rodent Diet; Lab Supply, Fort Worth, Texas). All procedures were approved by the Animal Care and Use Committee at Washington University in St Louis, the National Institutes of Health, Williams College, Virginia Tech and Monash University.

## Decision letter and Author response

Decision letter https://doi.org/10.7554/eLife.66173.sa1
Author response https://doi.org/10.7554/eLife.66173.sa2

## Additional files

### Supplementary files

• Transparent reporting form

### Data availability

The FED3 device is open-source and design files and code are freely available online at: https://github.com/KravitzLabDevices/FED3. In addition, we have made all data and analysis code for this paper available at https://osf.io/hwxgv/.

The following dataset was generated:

| Author(s) | Year | Dataset title | Dataset URL | Database and Identifier |
|---|---|---|---|---|
| Kravitz L, Matikainen-Ankney B | 2020 | Feeding Experimentation Device version 3 (FED3): An open-source device for measuring food intake and motivation in rodents | https://osf.io/hwxgv/ | Open Science Framework, hwxgv |

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
