## [Decision Letter]

**Acceptance summary:**

Thank you for submitting your work on FED3, a new and improved open-source option for a home cage pellet dispensing device, to *eLife*. The reviewers agreed that this open-source tool would be of wide-interest to neuroscience laboratories, that the manuscript was well-written and clear, and that the cross-lab validation was informative. They also appreciated that this Tools and Resource manuscript documents all necessary open-source hardware, firmware, visualization code, and Arduino and Python libraries for user customization of experiments and analysis.

**Decision letter after peer review:**

Thank you for submitting your article "Feeding Experimentation Device version 3 (FED3): An open-source device for measuring food intake and operant behavior" for consideration by *eLife*. Your article has been reviewed by 3 peer reviewers, one of whom is a member of our Board of Reviewing Editors, and the evaluation has been overseen by Kate Wassum as the Senior Editor. The following individual involved in review of your submission has agreed to reveal their identity: Daniel Aharoni (Reviewer #3).

Essential Revisions:

1. Describe in more detail the current commercial and open-source methods there are for monitoring food intake and operant behavior in rodent home cages. How does FED3 compare to those designs?

2. With the exception of panel 2D, little is shown to quantify jam-rate, which is one if the biggest issues with any pellet dispenser. Do the authors have any other data to this end? The authors mention in Figure 5D there is a retrieval time metric. Do you also have a time between pellet dispense onset and when the pellet dispenser detects that the pellet *has arrived* onto the pedestal?

3. In relation to the point "FED3 also has a programmable output that can control other equipment, for example to trigger optogenetic stimulation after a nose-poke or pellet removal, or to synchronize feeding behavior with electrophysiological or fiber photometry recordings." Do the requirements of this function impede one of the major strengths of this technology: large scale testing in laboratory vivariums? I would suspect tethers / wires in the home cage would be a major limitation to this feature for around the clock testing. Upon closer readings in the methods section, the mice were removed from the home cage for this part of the experiment. Perhaps the authors could make that clearer earlier on and separate describing this feature from the home-cage functionality of the device in its description.

4. For combining with optogenetic stimulation in Figure 7, is the FED3 output a single pulse trigger that is driving a pre-programmed train driven by another high-fidelity stimulation device or is the voltage train of 20hz being driven by FED3 directly turning on the LED? On a related note, how is the precision of timestamp recording for the external device? How is time stamp fidelity maintained across the session? It would be helpful to better understand how your system was set up to relate the time stamps of the data being written to the SD card and how well-registered they were after the fact.

5. Is FED3 water resistant? If a cage were to flood as sometimes happens with displaced water bottle tops, how well does the FED3 housing protect the electronics of the device and the animal?

6. Clarification or corrections needed for analysis/methods/statistics:

• Figure 3 shows an N of 10 but the manuscript text mentions you used 11 mice. This seems like a discrepancy in the paper.

• In the multi-site study of learning rates with FED3 section, you say "This highlights how FED3 enables high throughput studies of operant behavior and also demonstrates the potential for false positive effects when comparing between groups with small sample sizes (Figure 5B)". I feel like your claim that this data shows the potential for false positive effects is unsupported as I don't think you show that the individual groups must be pulled from the same distribution.

• For Figure 4E, please describe how the poke efficiency metric was calculated.

• In Figure 5 c – f, it is unclear if the error bars are across mice or across groups.

• In Figure 5 e and f, the vertical axis units are labeled as (%) but the vertical axis numbers look to be off by 100 times.

• In Figure 7 c, I think this is a plot of the cumulative poke count and not a temporally binned poke count. It would be nice to clarify this in the text or vertical axis label.

• Figure 7D shows data across 3 mice but, unlike most of your other figures, does not show the individual data points for each mouse. It would be nice to add those to the plot. It might be also worth considering showing the data from all 3 mice in figure 7C.

---

## [Author Response]

Essential Revisions:1. Describe in more detail the current commercial and open-source methods there are for monitoring food intake and operant behavior in rodent home cages. How does FED3 compare to those designs?

We have expanded our descriptions of common methods to track feeding, as well as commercial equipment used for rodent operant behavior in neuroscience labs. These expanded descriptions are in the first and second paragraphs of the introduction, marked in blue text.

2. With the exception of panel 2D, little is shown to quantify jam-rate, which is one if the biggest issues with any pellet dispenser. Do the authors have any other data to this end?

We thank the reviewers for raising this important point. We have not quantified jam-rate, in part because it has not been a major issue with FED3. Over the last 3 years we have made many tweaks and optimizations to the delivery chute to prevent pellet jamming. We now include a new supplemental figure (Supp. Figure 2) highlighting this chute design (see highlighted text in ‘Quantifying total food intake’). In our lab we have been running experiments with FED3 for >1 month in a closed economy setting with FED3 delivering thousands of pellets to each mouse over this time as their only source of food. An example showing this behavioral paradigm is also now included as a new figure in the manuscript (Figure 7).

The authors mention in Figure 5D there is a retrieval time metric. Do you also have a time between pellet dispense onset and when the pellet dispenser detects that the pellet has arrived onto the pedestal?

We thank the reviewers for bringing up this point. We empirically measured the latency time from nosepoke to pellet delivery to be ~2 seconds. We currently do not store the time between nosepoke and pellet arrival in the tray for each trial in FED3’s data file. This delay depends on how many motor movements FED3 takes to dispense a pellet, which for most trials is >5 as shown in the histogram in Author response image 1. While we decided not to store this information in the FED3 datafile, FED3 is open-source and we are happy to help any users calculate and store this information with their data if they find it necessary.

**Author response image 1. sa2fig1:** 

3. In relation to the point "FED3 also has a programmable output that can control other equipment, for example to trigger optogenetic stimulation after a nose-poke or pellet removal, or to synchronize feeding behavior with electrophysiological or fiber photometry recordings." Do the requirements of this function impede one of the major strengths of this technology: large scale testing in laboratory vivariums? I would suspect tethers / wires in the home cage would be a major limitation to this feature for around the clock testing. Upon closer readings in the methods section, the mice were removed from the home cage for this part of the experiment. Perhaps the authors could make that clearer earlier on and separate describing this feature from the home-cage functionality of the device in its description.

We thank the reviewers for noting this and have clarified that triggering external hardware is typically not done in a home cage (‘Discussion’ section, second paragraph). For example, self-stimulation of D1-MSNs (Figure 8) was conducted in a plastic arena outside the home cage. While FED3 was designed to be used in home-cages, we often use them attached to a plastic box for use as a replacement for an operant box. Using FED3 in this manner is much more cost effective than purchase of commercial operant boxes.

4. For combining with optogenetic stimulation in Figure 7, is the FED3 output a single pulse trigger that is driving a pre-programmed train driven by another high-fidelity stimulation device or is the voltage train of 20hz being driven by FED3 directly turning on the LED? On a related note, how is the precision of timestamp recording for the external device? How is time stamp fidelity maintained across the session? It would be helpful to better understand how your system was set up to relate the time stamps of the data being written to the SD card and how well-registered they were after the fact.

We thank the reviewers for asking us to clarify this. We have now clarified in the text that FED3 operates as a function generator to trigger LEDs or laser pulses (see ‘Optogenetic self-stimulation with FED3’). FED3 does not output enough current to drive high powered LEDs without an external power supply, but it can do all pulse timing on board.

In terms of precision of the ARM SAMD21 microcontroller that FED3 runs on, online documentation suggests that the GPIO pins on this chip can generate output pulses at up to 5MHz, achieving a precision of greater than 1 microsecond.

We do not have the equipment to test pulses this fast (and for most neuroscience applications this would never be needed). However, we attempted to empirically measure the precision of pulses generated by FED3 by sending a train of 20Hz pulses (Author response image 2, panel A) and recording the output of the BNC connector on an electrophysiology rig at 5 kHz; Here we were unable to measure any lag in the pulses (Author response image 2, panel B), so we conclude that the output pulses are at least as precise as 200 microseconds, and likely much more precise. We now mention this spec in the manuscript.

The reviewer also asks a different timing question in the second part of their comments. Here, they note that internal clocks can drift over time and ask how time-stamp fidelity is maintained across a session. We use a PCF8523 real-time clock (RTC) chip inside of FED3 to take care of this timekeeping. Therefore, the “drift” in timekeeping is limited by the precision of the PCF8523. We chose this RTC chip as it is cheap, easy to source, and can be controlled with a user friendly Arduino library written by Adafruit. However, this is not a high-precision RTC, it drifts up to 3 seconds per day. We considered this to be an acceptable drift for the intended use of FED3, which is to record food intake and operant behavior over a period of days or weeks. For most users, being ~1 minute off after a month of recording would be acceptable. We now mention this in the manuscript. In addition, should users need a higher precision real time clock, FED3 has an internal I2C auxiliary port that can accommodate any I2C peripheral. We are not aware of anyone yet using this port but we put it on the board in anticipation of future user needs. We now highlight this port in Figure 1D and mention it in the manuscript.

**Author response image 2. sa2fig2:** FED3 as a function generator. (A) Recorded 20 Hz Pulse train generated by FED3. (B) Peripulse rasters (top) and histogram showing change in voltage in 1ms around the onset of each pulse in a 20Hz train. Note that the observable lag in this test was zero. Y-axis is mV, x-axis in sec.

5. Is FED3 water resistant? If a cage were to flood as sometimes happens with displaced water bottle tops, how well does the FED3 housing protect the electronics of the device and the animal?

FED3 is not water-resistant, which we have clarified as a caveat (see the last paragraph of ‘Discussion’).

6. Clarification or corrections needed for analysis/methods/statistics:• Figure 3 shows an N of 10 but the manuscript text mentions you used 11 mice. This seems like a discrepancy in the paper.• In the multi-site study of learning rates with FED3 section, you say "This highlights how FED3 enables high throughput studies of operant behavior and also demonstrates the potential for false positive effects when comparing between groups with small sample sizes (Figure 5B)". I feel like your claim that this data shows the potential for false positive effects is unsupported as I don't think you show that the individual groups must be pulled from the same distribution.• For Figure 4E, please describe how the poke efficiency metric was calculated.• In Figure 5 c – f, it is unclear if the error bars are across mice or across groups.• In Figure 5 e and f, the vertical axis units are labeled as (%) but the vertical axis numbers look to be off by 100 times.• In Figure 7 c, I think this is a plot of the cumulative poke count and not a temporally binned poke count. It would be nice to clarify this in the text or vertical axis label.• Figure 7D shows data across 3 mice but, unlike most of your other figures, does not show the individual data points for each mouse. It would be nice to add those to the plot. It might be also worth considering showing the data from all 3 mice in figure 7C.

We have addressed these concerns throughout the manuscript, including expanding figure 8 (formerly figure 7) to show self-stim learning for all three mice. And we agree with the reviewers that while insufficient sample sizes can lead to false positive effects, we did not explicitly test this here. As such, we have clarified and scaled back our interpretation of the multi-site data shown in Figure 5.